# Hedgehog Signaling in Colorectal Cancer: All in the Stroma?

**DOI:** 10.3390/ijms22031025

**Published:** 2021-01-20

**Authors:** Natalie Geyer, Marco Gerling

**Affiliations:** 1Department of Biosciences and Nutrition, Karolinska Institutet, 14183 Huddinge, Sweden; natalie.geyer@ki.se; 2Theme Cancer, Oncology, Karolinska University Hospital, 17176 Solna, Sweden

**Keywords:** hedgehog signaling, colorectal cancer, tumor stroma

## Abstract

Hedgehog (Hh) signaling regulates intestinal development and homeostasis. The role of Hh signaling in cancer has been studied for many years; however, its role in colorectal cancer (CRC) remains controversial. It has become increasingly clear that the “canonical” Hh pathway, in which ligand binding to the receptor PTCH1 initiates a signaling cascade that culminates in the activation of the GLI transcription factors, is mainly organized in a paracrine manner, both in the healthy colon and in CRC. Such canonical Hh signals largely act as tumor suppressors. In addition, stromal Hh signaling has complex immunomodulatory effects in the intestine with a potential impact on carcinogenesis. In contrast, non-canonical Hh activation may have tumor-promoting roles in a subset of CRC tumor cells. In this review, we attempt to summarize the current knowledge of the Hh pathway in CRC, with a focus on the tumor-suppressive role of canonical Hh signaling in the stroma. Despite discouraging results from clinical trials using Hh inhibitors in CRC and other solid cancers, we argue that a more granular understanding of Hh signaling might allow the exploitation of this key morphogenic pathway for cancer therapy in the future.

## 1. Introduction

The Hedgehog (Hh) signaling pathway regulates organ development in both vertebrates and invertebrates and is indispensable for intestinal patterning. In this review, we summarize our current understanding of Hh signaling in colorectal cancer (CRC). First, we briefly describe the molecular basis of Hh signaling, from production and secretion of the ligand to the resulting target gene activation in the receiving cells. We then present the key cellular components involved in Hh-driven paracrine crosstalk between intestinal epithelial and stromal cells. Finally, we review the existing knowledge of Hh signaling in CRC and provide a perspective on how the multifaceted roles of Hh might be exploited for CRC therapy in the future.

## 2. Molecular Basis of Hedgehog Signaling

It has been forty years since the discovery of Hh signaling as a key regulator of developmental processes. In 1980, Nüsslein-Volhard and Wieschaus performed a mutational screen for embryonic lethality in *Drosophila melanogaster* [1]. They named the gene locus associated with a spiky appearance of the *Drosophila* larvae “hedgehog”. Fifteen years later, in 1995, both authors were awarded the Nobel Prize in Physiology or Medicine “for their discoveries concerning the genetic control of early embryonic development”, together with Edward B. Lewis [2].

### 2.1. Ligand Production and Secretion

In mammals, three Hh ligands exist, Indian Hedgehog (IHH), Sonic Hedgehog (SHH) and Desert Hedgehog (DHH). Hh ligands are produced as precursor proteins with two domains. During translation, the precursor translocates into the lumen of the endoplasmic reticulum (ER), and both domains undergo post-translational modifications. At the N-terminus, removal of a signal sequence reveals a conserved cysteine residue. Hh acetyltransferase (HHAT in vertebrates; skinny hedgehog [Ski] in *Drosophila*), which resides in the ER, recognizes this cysteine residue and catalyzes its palmitoylation. At the C-terminus, the Hh precursor is cleaved autocatalytically via its intein-like activity. First a protein-intein thioester is formed. Subsequently, cholesterol reacts with the thioester, the precursor protein is cleaved, and the former C-terminus is degraded within the ER. As a result, the remaining Hh protein is modified with two lipid structures, a palmitoylated N-terminus and a cholesterol-conjugated C-terminus [3] (Figure 1a).

Due to their lipid moieties, Hh ligands are hydrophobic; thus, in order to convey long-range signals through a tissue, several proteins are required to aid in signal transduction to distant cells. One of them is the transporter-like protein Dispatched1 (DISP1, reviewed in [4]). Loss of DISP1 leads to the accumulation of Hh ligands on the surface of producing cells. As a result, long-range signaling is impaired, while activity in adjacent recipient cells remains intact [5,6]. DISP1 is a membrane transporter composed of a 12-pass transmembrane domain and two extracellular domains [7], and is essential for correct SHH secretion together with its cofactor, secreted signal peptide, CUB and epidermal growth factor-like domain-containing protein 2 (SCUBE2). Recently, a model has been proposed in which SCUBE2 binds SHH and transports it towards the recipient cell [8].

Heparan sulfate proteoglycans (HSPGs) are components of the cell surface and extracellular matrix that are composed of a protein, such as syndecan or glypican, with attached heparan sulfate chains. Syndecans are transmembrane proteins, and HSPGs with membrane-anchored glypicans are found on the outer surface of the cell membrane [9,10]. Glypicans can be released from the membrane and play a crucial role in long-range ligand transportation of several morphogens, including Hh and Wnt ligands [10,11]. Specifically, impaired assembly of glypican HSPGs, either by loss-of-function of the proteins themselves (dally and dally-like are important glypicans in *Drosophila*, glypican 5 in vertebrates) or by loss-of-function of the glycosyltransferases (encoded by the genes, *tout-velu* [*ttv*] and *brother of tout-velu* [*botv*] in *Drosophila*, exostosin 1 and 2 [*Ext1* and *Ext2*] in vertebrates), interferes with Hh long-range signaling and results in the accumulation of Hh ligands in close proximity to the producing cell [12,13,14]. Glypicans and syndecans are mainly produced by intestinal epithelial cells [9,15], whereas perlecan, a secreted HSPG, is the dominant HSPG in the extracellular matrix [15].

Based on this molecular machinery, cells that secrete Hh ligands can initiate ligand-dependent signals in a cell autonomous manner (autocrine), in directly adjacent cells (juxtacrine), and in distant cells (paracrine).

### 2.2. Reception of the Ligand

On the recipient cell, the two transmembrane proteins, patched 1 (PTCH1) and smoothened (SMO), are the main transducers of ligand stimuli (reviewed in [16]). In the absence of a ligand, the Hh receptor PTCH1 represses SMO function by preventing it from entering the primary cilium (Figure 1b), an antenna-like structure found in most mammalian cells. However, intestinal epithelial cells lack a primary cilium, while ciliated cells are found in the intestinal mesenchyme [17,18]. Binding of Hh ligands to PTCH1 releases its repression of SMO and results in the accumulation and activation of SMO in the primary cilium (Figure 1c).

Although the mechanisms of interaction between PTCH1 and SMO are not fully understood, recent studies have revealed the importance of a cholesterol transporter function for PTCH1 through both leaflets of the plasma membrane bilayer (reviewed in [16]). In its active form, the transmembrane protein maintains low levels of cholesterol in the inner leaflet. Inactivation by SHH ligand binding via its palmitoylated N-terminus to the extracellular domain of PTCH1 induces a conformational shift of the extracellular protein domains, locking PTCH1 in a conformation that makes it incapable of transporting cholesterol through the plasma membrane. As a consequence, ligand binding to PTCH1 increases the local concentration of cholesterol in the inner leaflet of the plasma membrane [19,20].

SMO is a class F G-protein-coupled receptor, which is activated by cholesterol binding to its extracellular cysteine rich domain (CRD) [21,22]. Crystallization of active SMO has recently revealed that SMO activation leads to conformational changes of the CRD and transmembrane domains, transducing downstream signaling. Interestingly, active SMO additionally forms a tunnel with sterol-binding sites within the transmembrane domain [23,24]. This observation suggests a mechanism by which SMO channels cholesterol from the inner leaflet of the plasma membrane towards the extracellular space, where it binds to the CRD and activates downstream signaling. Recently, it has been shown that SMO is actively removed from the primary cilium by ubiquitination and subsequent degradation [25]. Multiple epidermal growth factor-like domains 8 (MEGF8) and mahogunin ring finger-1 (MGRN1) regulate SMO ubiquitination. Loss-of-function in either of these proteins induces accumulation of SMO in the primary cilium and increases cellular responsiveness to SHH, resulting in the loss of a graded transcriptional response [25,26]. Thus, MEGF8 and MGRN1 are examples of modifiers of the Hh downstream response, which is not only dependent on its core “canonical” components, but is subject to external influences, many of which are still elusive.

Aside from PTCH1, Hh ligands bind to three known coreceptors, growth arrest-specific 1 (GAS1), CAM-related/downregulated by oncogenes (CDO), and brother of CDO (BOC) [27]. Several studies suggest tissue-specific roles for these coreceptors in modifying Hh signaling during embryonic development. Simultaneous loss-of-function of GAS1, CDO (also known as CDON), and BOC proteins causes severe developmental malformations, partly resembling SHH loss-of-function phenotypes [28,29,30,31]. A recent study suggested that GAS1 interacts with PTCH1 in the absence of SHH and in the early stages of pathway activation, but dissociates from PTCH1 during long-term pathway stimulation, suggesting a role in the modification of immediate, but not continuous signals [32]. In addition, CDO and BOC play crucial roles in retaining the complex of SHH and its extracellular transport protein SCUBE2 at the cell surface [8].

### 2.3. Downstream Signal Transduction

Within the recipient cell, the transcription factors, GLI1, GLI2, and GLI3 are the main downstream executors of Hh activation. GLI2 and GLI3, but not GLI1, contain an N-terminal repressor domain and can function as both transcriptional repressors and activators [33,34]. In the absence of upstream pathway activation (Figure 1b), GLI2 and GLI3 are phosphorylated by protein kinase A (PKA) and converted into their truncated repressor forms, which enter the nucleus and inhibit the expression of target genes. In addition, the negative pathway regulator, suppressor of fused (SUFU), forms complexes with the GLI transcription factors [35]. SUFU is essential for mammalian Hh signaling transduction, as demonstrated by the early embryonic lethality of both *Sufu^-/-^* mice and mice with a hypomorphic *Sufu* allele [35,36]. SUFU binds GLI proteins intracellularly, whereby a conformational change of an intrinsically disordered region of the SUFU protein can be observed [37]. Functionally, SUFU may act as a chaperone of the GLI proteins, influencing their nuclear translocation [38], although the complex consequences of SUFU–GLI interaction on transcriptional output are still incompletely understood.

Activation of SMO (Figure 1c) results in the inactivation of the two negative pathway regulators, PKA and SUFU, within the primary cilium. As a consequence, the PKA-mediated phosphorylation of GLI2/3 is inhibited, resulting in reduced proteolysis. Additionally, full-length GLI2/3 proteins dissociate from SUFU and are phosphorylated by ULK3 and STK36, mammalian homologues of Fused (Fu), a *Drosophila* serine/threonine kinase [34], leading to the formation of GLI2/3 activator forms. Graded dephosphorylation at PKA target sites and increasing phosphorylation at ULK3/STK36 target sites enhances the transcriptional activity of GLI2/3 [33,34]. As such, GLI2/3 phosphorylation patterns act as regulators that fine-tune Hh pathway activity. Once activated, GLI2 and/or GLI3 translocate to the nucleus and induce the expression of Hh target genes. The major target gene, *GLI1*, which acts solely as a transcriptional activator, serves as a signal amplifier in a positive feedback loop.

Two genome-wide CRISPR-screens on mouse fibroblasts with Hh-responsive reporter systems have identified positive and negative regulators of Hh pathway activity [26,39]. The results from both screens included classical Hh components such as the positive regulator GLI1, and the negative regulators, PTCH1, GLI3, SUFU. The majority of the remaining regulators of Hh signaling identified were associated with ciliogenesis. Interestingly, two novel factors regulating ciliogenesis, FAM92A and TTC23, have been discovered and functionally validated [39] and the impact of the SMO negative regulatory proteins, MEGF8 and MGRN1, was underscored by the data [26].

The three mammalian transcription factors, GLI1–3, all contain five conserved zinc finger DNA binding domains that allow them to interact with a consensus GLI-binding motif found in promoter or enhancer regions of Hh target genes [40,41,42]. While *GLI1* and *PTCH1* are generally considered the most robust target genes, for which upregulation can be expected in essentially all cell types with an active Hh downstream machinery, other targets are dependent on cell type and cellular context, as well as on the balance of GLI repressor and activator levels (reviewed in [43]). Important examples of pathway convergence with Hh signaling are epithelial growth factor (EGF) and fibroblast growth factor (FGF) signaling [44,45], both of which can modify the downstream output of Hh activation. Thus, the consequences of Hh ligand binding are not only dependent on the amount of bound ligand, but are also modified by a multitude of known and unknown factors, such as the protein levels of the receptors, PTCH1 and SMO, and the coreceptors (such as GAS1, CDO and BOC), as well as intracellular modifiers, which are regulated by several external, independent pathways.

### 2.4. Canonical vs. Non-Canonical Hedgehog Signaling

Activation of target genes via Hh ligand binding to PTCH1, which then activates GLI-mediated transcription via SMO, is referred to as “canonical” Hh signaling. Alternatively, GLI transcription factors can be activated independently of PTCH1, SMO, or both, which is then referred to as “non-canonical” Hh signaling (we kindly refer the reader to a recent review on non-canonical Hh signaling [46]). The list of potential signaling pathways that can induce non-canonical GLI1 activation is long and includes RAS-RAF signaling, the p53 pathway, and the transforming growth factor (TGF) beta pathway, as well as other signaling pathways that are of known importance for oncogenesis in general, and colorectal carcinogenesis in particular.

In summary, canonical Hh signaling depends on ligand production, autocrine, juxtacrine or paracrine reception of the ligand, and on a complex intracellular signaling transduction machinery that underlies influences from converging and inhibiting pathways. In addition, downstream effector activation can be non-canonical, and as such can become uncoupled from ligand production. The main practical readout of canonical Hh downstream activity is the level of the transcriptional activator *GLI1*.

In the following section, we will first discuss signal direction and downstream targets of canonical Hh signaling in the intestine during development and homeostasis, before summarizing the current evidence for the role of Hh signaling in CRC.

## 3. Intestinal Hedgehog Signaling in Development and Homeostasis

### 3.1. Importance of the Intestinal Stroma for Epithelial Maintenance

Both the small intestine and the colon are highly proliferative tissues. Their epithelium consists of a monolayer with a well-defined cellular hierarchy that enables constant cell renewal. Stem cells that depend on high levels of Wnt signaling are located in the crypts of both small intestine and colon [47]. In the small intestine, these epithelial stem cells are maintained by essential Wnt ligands secreted by both neighboring epithelial Paneth cells [48] and adjacent stromal cells [49]. In the colon, Wnt ligands are produced exclusively by subepithelial mesenchymal cells [49]. Along the differentiation trajectory towards the lumen, from where cells are shed within 4–5 days of their emergence, bone morphogenetic protein (BMP) signaling is the major inducer of differentiation. Both BMP ligands, for example, BMP4 or BMP5, and inhibitors, such as Noggin or Gremlin-1 (GREM1) are secreted mainly by the adjacent stromal cells rather than by epithelial cells [50,51]. Hence, the intestinal stroma can be considered as the conductor of epithelial homeostasis, using morphogenic signaling pathways as its baton.

### 3.2. The Role of Hedgehog Signaling for Intestinal Development: A Paracrine Requirement

The importance of the Hh pathway for intestinal development in mammals was initially shown using *Shh* and *Ihh* knockout mice, which display a wide range of intestinal malformations [52,53]: knockout of *Shh* led to foregut malformations such as duodenal stenosis or a malformed esophagus, and to disturbed intestinal innervation [52,53]. The *Ihh* knockout was associated with impaired differentiation of intestinal epithelial cells, and with the loss of enteric ganglions, reminiscent of Hirschsprung’s disease in humans, in which nerve ganglions are lacking in parts of the colon at birth [52]. In both cases, a common consequence of the loss of either *Ihh* or *Shh* was the reduction of the intestinal smooth muscle layer [52]. Similar gastrointestinal phenotypes were observed upon deletion of either of the Gli transcription factor genes, *Gli2* and *Gli3* [54], while a *Gli1* knockout alone did not induce detectable changes in the developing gut [55].

It is now a generally accepted model that Hh signaling in the small intestine and colon is predominantly paracrine in both development and homeostasis: epithelial cells secrete the Hh ligand, which is received by adjacent mesenchymal cells. In a comprehensive study addressing the direction of Hh signals in the developing mouse gut, genetically engineered reporter mice that indicate expression of the downstream effectors, *Gli1*, *Gli2*, *Ptch1*, as well as the ligand *Shh*, were combined with in situ hybridization studies to show that Hh signaling is strictly paracrine from the gastric antrum to the anus [56]. Hh-responsive cells, as defined by the expression of *Ptch1* and *Gli2* were located subepithelially in the cores of the villi of the small intestine, and in the external smooth muscle layer, while epithelial cells expressed *Ihh* [56]. Functionally, overexpression of *Ihh* in epithelial cells increased the abundance of smooth muscle cells in the cores of the villi [56]. The finding that intestinal Hh signaling is always and strictly paracrine during development [56] contrasts with an earlier report that proposed a role for intra-epithelial Hh signaling in the maintenance of Paneth cell homeostasis. In this study, mature Paneth cells of the small intestine were found to express Ihh, which negatively regulated their intraepithelial precursors expressing both *Ptch1* and Hedgehog interacting protein (*Hhip*), another target gene [57].

A particularly good illustration of the paracrine wiring of Hh signaling stems from mechanical studies on chicken intestine: In the developing, flat epithelial monolayer of the chicken gut, intestinal stem cells and differentiated cells are evenly distributed. The intermingled differentiated cells secrete the Hh ligand, SHH. When the monolayer bends to form a villous-like structure on top of its endogenous mesenchyme, accumulation of the Hh ligand in the stromal layer induces BMP4 expression, which in turn leads to differentiation of the overlying epithelial cells, directing the intestinal stem cells away from the apical part of the villus and towards the crypt [58]. While the positioning of stem and differentiated cells can be a direct consequence of intestinal folding due to smooth muscle tension in chickens [58], a more complex mechanism might be at work in the mammalian intestine [50]: in the absence of mechanically-induced epithelial folding in mice, complex clusters of BMP ligands and modulators, such as the secreted BMP inhibitor Noggin, orchestrate villi emergence [50]. Interestingly, clusters of mesenchymal cells with active Hh signaling aggregate during this process and determine the size of the villi [59].

In both chickens and mice, active Hh signaling in the stroma has pro-differentiating effects on the overlying epithelial layer, such that stromal Hh activation leads to the loss of epithelial progenitors [58,60,61], while reduced Hh signaling evokes an expansion of the epithelial stem cell compartment associated with enhanced Wnt signaling [62,63,64]. There are indications from chronic overexpression or deletion of *Ihh* in the small intestine, that these effects are most pronounced in the short-term, and that modifications of Hh ligand abundance can be partly compensated for over time [65,66].

### 3.3. Response to Hh Ligands: Cell Types and Activated Target Genes

Both the direction of Hh signaling in the intestine—from the epithelium to the stroma—and its pro-differentiating net effect on the epithelium are well established (Figure 2). However, less is known about: (i) the different stromal cell types involved in the response to Hh ligands, and (ii) the target genes under the control of the canonical Hh pathway in the intestine. Two recent studies have shed light on these questions. First, intestinal stroma cells expressing *Gli1*, the classical target indicating active downstream Hh signaling, were identified as the source of the Wnt ligand, WNT2B, which is essential for maintenance of the intestinal stem cell state [67]. Focusing specifically on the colon, the same group went on to show that Gli1-cell specific genetic deletion of Wntless (*Wls*), a key factor for Wnt protein secretion, led to complete destruction of the overlying epithelium, accompanied by the loss of Wnt downstream targets [49]. These results indicated that at least some *Gli1+* stromal cells provide essential Wnt ligands to colonic stem cells, and that the dependency on Wnt ligands could not be rescued by WNTs from other stromal cells not expressing GLI1. Interestingly, single cell RNA sequencing revealed eight distinct clusters of *Gli1+* stromal cells in the mouse colon, two of which were enriched for the Wnt ligands, WNT2, WNT2B, and WNT4, which the authors could spatially map to the bottom of the crypt. Thus, colonic *Gli1+* cells are a highly heterogeneous cell population, and only a subset of *Gli1+* cells expresses Wnt ligands. These findings make it possible to reconcile the data showing that *Gli1+* cells are essential providers of Wnt ligands in the colon with the overall net effect observed in other models, in which stromal Hh activation leads to increased differentiation of epithelial cells, rather than expansion of the epithelial stem cell compartment [60,62]. The heterogeneity of *Gli1+* cells could also be important for therapies directed at manipulating Hh signaling in cancer, as this might have diverse effects that depend on the specific cell type targeted and its spatial location with respect to tumor cells, as we will discuss below.

While Wnt ligands are expressed by *Gli1+* cells in the colon [49], it is not clear if they are the direct target genes of canonical Hh signaling. Several earlier studies that have used mouse models or in vitro systems to decipher Hh target genes have identified sets of genes with diverse biological functions that potentially lie downstream of canonical Hh signaling in the receptive stromal cells. The variation in the results from these studies could be a consequence of the cellular heterogeneity of *Gli1+* cell populations. Furthermore, difficulties in defining the exact targets of Hh in the intestinal stroma indicate that these target genes might be highly cell-type and/or context-dependent. In one of the first studies to explore intestinal Hh targets beyond development, mouse embryonic mesenchyme was treated ex vivo with the Hh ligands, IHH and SHH [65]. Ligand treatment for 24 h robustly induced Myocardin (*Myocd*), in parallel with upregulation of the known Hh targets, *Gli1* and *Ptch1*. *Myocd* is a master regulator of smooth muscle cell differentiation, offering a possible mechanism for how Hh controls mesenchymal cell number and differentiation in intestinal homeostasis. In addition to *Myocd*, upregulation of the BMP agonist, *Bmp4*, was observed. BMPs induce differentiation of intestinal epithelial cells [69], hence BMP4 expression in the underlying stroma is expected to induce differentiation of adjacent epithelial cells. The same group used microarray gene expression analysis of similarly treated, isolated murine intestinal mesenchyme [70]. As expected, *Ptch1* and *Gli1* were upregulated after both IHH and SHH treatment, alongside a total of 27 upregulated and 75 downregulated genes. These regulated genes represent diverse biological functions; as expected, most had previously been associated with organ development as well as cell differentiation and proliferation, while a significant number of transcripts had a known role in regulating immune cell functions, including Interleukin 6 (*Il6*), and several chemokines [70]. However, *Bmp4* was not included in the list of differentially regulated genes in this dataset, neither were Wnt ligands. Importantly, the authors also showed that culturing intestinal mesenchyme, as was done prior to treatment with Hh ligand, induced several key inflammatory pathways, suggesting that canonical Hh signaling can function as an anti-inflammatory signal acting via the mesenchyme [70]. Indeed, an anti-inflammatory role for stromal Hh signaling has been proposed in several other studies [62,71,72,73], although the exact mechanisms by which the stromal response to epithelial Hh ligand modifies the immune response are only partly understood [74]. While it has been suggested that inflammatory cells from the myeloid lineage actively respond to Hh and upregulate *Gli1* [70,73], a recent comprehensive study based on genetically engineered mouse models found no evidence for canonical Hh activity in intestinal immune cell lineages [72], suggesting that the effect of Hh ligands on inflammatory responses does indeed require a stromal, non-inflammatory cell intermediary.

In addition to BMP4, which was upregulated in intestinal mesenchyme upon Hh ligand treatment ex vivo [65], other BMP agonists, including BMP2, BMP5, and BMP7 are induced in the stroma upon activation of canonical Hh signaling [50,60,75]. Furthermore, findings in mouse models suggest that BMP inhibitors such as *Grem1* or twisted gastrulation BMP signaling modulator 1 (*Twsg1*) are controlled by Hh signaling, although data on the direction of the regulation are inconsistent [61,75]. Intestinal BMP ligands and antagonists are largely secreted by stromal cells [76,77], and they form a gradient with the antagonists found most abundantly at the crypt bottom, and agonists closer to luminal enterocytes [78].

While Hh appears to control the expression of both BMP agonists and antagonists in the stroma, the net effect of experimentally reduced Hh activity in the mouse intestine is a decrease in epithelial BMP activity, as assessed by reduced expression of the BMP transcriptional targets, Inhibitor of DNA binding 1 *(ID1), ID2*, and *ID4*, as well as by a decrease in the abundance of phosphorylated SMAD1/5 (mothers against decapentaplegic homolog 1/5), receptor kinases regulated by BMP signaling [75]. Consistently, the activation of Hh signaling in the mouse intestine by conditional genetic loss of functional *Ptch1* led to the increased expression of *ID2* and *ID4*, as well as more abundant phosphorylated SMAD1/5 in epithelial cells [60]. Increased Hh activity also reduced Wnt activity in the intestinal crypt and caused crypt hypoplasia, linking increased stromal Hh activation to reduced Wnt activity [60].

Forkhead box (FOX) transcription factors are evolutionarily conserved regulators of development and homeostasis in various organs [79]. The members of the Fox gene family, *Foxf1* and *Foxl1*, are direct targets of the GLI proteins and mediators of stromal Hh signaling in the developing intestine [63]. Recent studies have provided insights into the identity of the stromal cell type at the center of the Hh–FOX axis; *Foxl1+* stromal cells are rare and possess cytoplasmic extensions that stretch across hundreds of micrometers [49,80], possibly enabling contact with tens of overlying epithelial cells (reviewed in [80]). These cells are now most frequently referred to as telocytes and their cellular protrusions are termed telopodes [80]. In contrast to other mesenchymal cells, such as myofibroblasts, telocytes produce the Wnt ligands and BMP antagonists required for intestinal stem cell maintenance. Furthermore, recent data suggest that telocytes are able to compartmentalize the expression of signaling molecules, such that they secrete stem cell factors near the crypt base, while the same cell can secrete other factors in regions closer to the lumen [81]. It will be interesting to understand in detail whether Hh signaling is needed for (i) the induction and maintenance of telocyte identity, (ii) the expression of specific signaling molecules, and (iii) the compartmentalization of signaling molecule secretion across the crypt.

### 3.4. Hedgehog I s Critical for Epithelial Regeneration

Experiments that put stress on the adult intestinal epithelium in mice have consistently revealed a critical role for Hh signaling in the maintenance of intestinal homeostasis. Mice treated with dextran sulfate sodium (DSS), a sugar that induces epithelial cell damage in the colon and subsequently leads to intestinal inflammation resembling important aspects of human ulcerative colitis, depend on a functional Hh pathway for tissue regeneration: when mice lacking one or both *Gli1* alleles were challenged with DSS, intestinal inflammation was more severe than in wild type control mice [71,73]. Mice with epithelial-cell loss of *Ihh* or stromal loss of *Smo* were more susceptible to DSS, and treatment with an Hh agonist ameliorated the effect of DSS treatment [71]. The mechanisms behind these effects are not entirely understood, although induction of protective IL-10 [71], or suppression of pro-inflammatory CXC-motif-chemokine 12 (CXCL12) [72] have been suggested as potentially important mediators of the protective role of Hh signaling in the context of mucosal injury and inflammation.

In summary, canonical Hh signaling in the developing intestine relies on ligand production by epithelial cells, while its downstream effects are mediated by adjacent stromal cells. The stromal cell types receptive for the Hh ligand are diverse and likely comprise both cells located at the crypt bottom and those towards the crypt apex. Importantly, these mesenchymal cells have opposing functional roles, such that stromal cells are required for maintenance of the epithelial stem cell compartment in the crypt base, whereas stromal cells located towards the intestinal lumen secrete factors inducing epithelial differentiation. While stromal Hh pathway activity and epithelial differentiation are intimately linked, the Hh pathway lies at the heart of additional important processes including the regulation of the intestinal immune response and intestinal smooth muscle homeostasis.

## 4. Stromal Hedgehog Signaling in Colorectal Cancer

### 4.1. Epidemiology and Genetic Background of Colorectal Cancer

Together with lung, breast, and prostate cancer, CRC is one of the four most common cancer types in adults worldwide [82]. One person in 25 will be diagnosed with CRC during their lifetime [82], and in the United States alone, CRC causes more than 50,000 deaths per year [82]. Most CRC cases are sporadic, i.e., they arise in an otherwise healthy colon devoid of known underlying conditions that cause predisposal to cancer development. The risk of developing a colorectal tumor increases with age, and multiple factors contribute to its pathogenesis; risk factors include a high intake of red meat, low physical activity, cigarette smoking, inflammatory bowel disease, and hereditary diseases [83]. More recently, the contribution of specific intestinal bacteria to colonic carcinogenesis has become clearer, as it has been shown that colibactin, produced by *Escherichia coli*, induces a distinct mutational signature that can be identified in a subset of CRCs [84].

The most important genetic disorders that confer an increased risk for CRC are hereditary non-polyposis colorectal cancer (HNPCC) [85], in which mutated genes for DNA mismatch repair result in a greatly increased CRC risk, and familial adenomatous polyposis (FAP), which in most cases is caused by mutations in the Wnt pathway gene, adenomatous polyposis coli(*APC*), leading to the development of hundreds or thousands of intestinal polyps with a high risk of progression to CRC [86].

In the majority of sporadic colorectal tumors, overactivated Wnt signaling [87,88] acts as the key oncogenic driver. Most frequently, Wnt activation in CRC is caused by genetic alterations downstream in the Wnt signaling cascade [89]. Of the genes involved in the Wnt pathway, the tumor suppressor, *APC*, is the most frequently mutated. Genetic alterations in other Wnt genes such as *CTNNB1* (encoding for beta-catenin) [90], or fusions involving R-spondin family members that potentiate Wnt signals, play a role in a smaller proportion of cases [91]. Other clinically important genetic alterations in CRC include activating mutations in *KRAS* and *BRAF*, both of which are used for therapeutic decision making in clinical practice [92,93], while less frequently, mutations in *NRAS* are found [94]. Furthermore, a defective DNA mismatch repair system (d-MMR), resulting in microsatellite instability (MSI), characterizes approximately 25% of all CRCs and is associated with a favorable response to checkpoint inhibitors [95]. On a transcriptional level, CRC can be classified into four Consensus Molecular Subtypes (CMS), such that CMS1 corresponds to MSI tumors with a strong immune response, CMS2 comprises tumors with high Wnt activation (“canonical” CRC), CMS3 is characterized by metabolic dysregulation, and CMS4 by activated stroma [96].

### 4.2. Wnt Signaling as the Major Oncogenic Pathway in Colorectal Cancer

Despite the importance of other genetic alterations that modify tumor aggression, Wnt signaling is the primary tumor cell-intrinsic pathway that drives CRC oncogenesis. Wnts act as strong mitogens for intestinal epithelial cells, as was demonstrated strikingly in mice treated with the Wnt antagonist, Dickkopf-1 (DKK1), in which proliferation in the intestinal crypts halted within days after treatment, leading to crypt degeneration [97]. Interactions between Wnt and Hh signaling in CRC at different stages of tumor development are of interest for at least two reasons: firstly, stem cells are more susceptible to malignant transformation than differentiating cells located closer to the lumen [98]. The intestinal stem cell state is particularly dependent on the fine-tuned activation of epithelial Wnt signaling [48,67]. Mouse models suggest that cells further downstream on the differentiation trajectory (closer to the lumen) can act as tumor-initiating cells only when “pushed back” towards the stem cell state [77,99]. Secondly, Wnt target genes are almost uniformly upregulated in CRC [89], and *LGR5+* colon cancer cells with particularly high Wnt activity reside at the top of a cellular hierarchy in human tumors, giving rise to high numbers of differentiated progeny [100]. *LGR5+* cells play a particularly important role in the maintenance of CRC liver metastases [101]. Extrapolating from its role as a strong regulator of the intestinal stem cell state in development and homeostasis [56,60,62,63,102,103], it is likely that stromal Hh signaling interacts with epithelial Wnt signaling to influence both the pool of stem cells susceptible to malignant transformation and the cancer stem cell state in the tumor cell hierarchy. In non-malignant colonic mucosa, stem cells are maintained by external Wnt ligands, which in the colon are secreted by adjacent stromal cells [49,67,104]. As outlined above, some of these cells in the crypt base also express the major Hh target GLI1, while most *Gli1+* cells do not express Wnt ligands [49]. Activation of the Wnt pathway leads to intestinal tumors in mice, and a plethora of mouse models has been developed based on this finding [105,106,107]. Importantly, attenuating Wnt signaling by restoring *Apc* function in mice with established colonic tumors bearing additional mutations in *Kras* and *p53*, led to tumor regression, suggesting that high Wnt activity is not only essential for tumor initiation, but is also required for tumor maintenance and progression [108].

### 4.3. Crosstalk between Signaling Pathways in Colorectal Cancer

In CRC, Wnt-activating mutations act cell-autonomously, and hence a major question is “To what extent do cancer cells remain susceptible to external modifications of Wnt signaling and other interacting pathways such as stromal Hh signaling?” Functional data on the role of Hh signaling in CRC from mouse models that address this question are contradictory. Our group has previously shown that stromal activation of canonical Hh signaling by genetic loss of *Ptch1* in colonic stromal cells can attenuate tumor growth in a chemically induced colorectal tumor model. This effect was partly mediated by the modification of BMP inhibitor expression, such that *Grem1* was upregulated upon inhibition of Hh signaling, and downregulated upon its activation [61]. In line with this finding, knockout of *Ihh* from the epithelium resulted in reduced stromal Hh activity and an increased tumor burden, as did treatment with a SMO inhibitor. Similar results were observed independently by another group using a different genetic model to modify Hh signaling: in a colitis-associated colon tumor model, mice with whole-body loss of one *Ptch1* allele showed a reduction in the number and size of colorectal polyps compared to wild type mice [71]. Mechanistically, the authors found that IL-10 was induced by stromal Hh activation, which had a protective effect on the course of colitis. In the chemical model of colonic tumorigenesis used in both studies, [61,71], tumors are induced by administration of the mutagenic agent, azoxymethane (AOM), followed by repeated cycles of DSS to induce epithelial damage and subsequent colitis, which accelerates tumor development [109]. The severity of colonic inflammation (or epithelial destruction) is directly correlated to the number and size of the resulting tumors [109], making it difficult to untangle influences of the inflammatory response to DSS-induced injury and more direct effects of stroma–tumor crosstalk. To address this, the authors used inducible *Gli1Cre^ERT2^; Smo^fl/fl^* mice and treated first with AOM/DSS, and then, after the last cycle of DSS, gave tamoxifen to remove *Smo* from *Gli1*-expressing cells. Compared to *Gli1Cre^ERT2^;Smo^fl+^* mice that retained one functional *Smo* allele and hence the capacity to activate Hh signaling, mice with complete loss of *Smo* in *Gli1*-expressing cells had an increased tumor burden [71]. We used the opposite approach based on *Col1a2Cre^ER^; Ptch1^fl/fl^* mice, in which stromal Hh signaling is activated upon Tamoxifen administration. We induced tumors with AOM/DSS, then established the baseline tumor volume as measured in vivo by ultrasound, followed by the activation of Hh signaling by administration of tamoxifen. We observed tumor growth arrest and the regression of some tumors, comparable in quality to the effect of *Apc* restoration [108]. We found that *Grem1* expression was reduced in the stroma of tumors from mice with Hh activation.

Interestingly, in *Apc^min^* mice, another widely used model of intestinal adenomas, epithelial *Ihh* was required for tumorigenesis [66]. These contradictory results can possibly be explained by the significantly different requirements of stromal signals for epithelial homeostasis in the small intestine and colon [49,104], or by differences in the mutational background of the two models [110]; in any case, they highlight the necessity to relate all functional data from animal models to human CRC.

### 4.4. Epithelial and Non-Canonical Hedgehog Signaling in Colorectal Cancer

In contrast to its paracrine role in the colon, tumor cell-intrinsic activation of Hh signaling caused by genetic alterations characterizes most basal cell carcinomas (BCCs) and is the cause of nevoid basal cell carcinoma syndrome (NBCCS or “Gorlin syndrome”) [111,112]. In the 1990s and following decades, this finding sparked a huge interest in possible cell-autonomous oncogenic roles for Hh in solid cancers. In a systematic study of Hh target gene expression in digestive tract cancers, Berman et al. reported widespread activation of Hh signaling in cancers of the gastrointestinal and pancreaticobiliary tract, but—notably—not in CRC, despite the upregulation of *SHH* [113]. Importantly, activation of Hh in these tumor types was not due to mutations, but was a consequence of increased ligand secretion [113]. An analysis of CRC cell lines found that the most commonly used lines do not express Hh downstream targets [114]. A study using immunohistochemistry for GLI1 also found downregulation of Hh signaling in CRC compared to normal mucosa, and described an inhibitory, cell-intrinsic effect of experimental GLI1 overexpression on Wnt activity [115]. Data from another group supported the tumor cell-intrinsic activation of Hh signaling: when enriching for *CD133^+^* as a marker for cancer stem cells [116], this study reported increased *GLI1* and *HHIP* expression in CRC cancer stem cells compared to normal mucosa [117]. The latter study, conversely, found a dependency of CRC cancer stem cells on active Hh signaling [[117] and reviewed in [118]].

In an analysis of gene expression data from The Cancer Genome Atlas (TCGA), we have previously shown that *SHH* is indeed upregulated in CRC compared to non-malignant colonic mucosa [61]. However, *IHH*, which is the main intestinal ligand, is downregulated, along with the canonical Hh targets, *GLI1* and *HHIP* [61], supporting the study by Berman et al. [113]. Despite the inherent limitations of bulk gene expression data analysis, including a lack of information as to which cells express which transcripts, it can be concluded that increased *SHH* expression does not translate to an increased overall expression of Hh target genes in CRC.

The data on the expression of Hh components in colonic adenomas, precursors of CRC, are less clear. Initially, it was reported that IHH protein expression was already lost in dysplastic mucosa [103], and that SHH expression was increased [119]. However, a more recent study found also *IHH* to be upregulated in adenomas on the mRNA level [66].

Zooming in on the different CRC cell compartments such as tumor cells, fibroblasts and immune cells, based on sorted cell populations [120], we found that Hh target genes are expressed almost exclusively by non-immune stromal cells of the tumor microenvironment, while ligand expression is restricted to tumor cells, as expected [61]. These data are supported by studies on reporter mice used to visualize *Gli1*-expressing cells and their progeny: in *Apc^min^* mice that develop small intestinal tumors and in mice with chemically-induced colorectal polyps, *Gli1^+^* cells reside exclusively in the stroma, whereas epithelial activity was never observed [61,66]. Hence, there are strong data to support the idea that Hh activity in CRC is compartmentalized to the stroma, rather than to the epithelium. However, the finding of epithelial Hh activity in some studies [117,121], could still point to a role for Hh downstream targets in either a subgroup of patients or a subset of specialized and relatively rare colon cancer cells that may depend on Hh signaling. Indeed, independent studies have suggested a role for non-canonical, cell-intrinsic Hh signaling in colon cancer cells. In their analysis of label-retaining or “dormant” cancer cells from an *Apc-*mutated mouse model, one group reported this particular cancer cell state as characterized by high levels of Hh signaling [122]. However, inhibition of GLI1 with the GLI antagonist, GANT61 [123], or treatment with Hh ligands had no effect on CRC cell lines. Instead, the crosstalk between Hh and Wnt was mediated by SUFU, acting downstream of SMO [122]. Another study found that CRC cancer stem cells, defined by high expression of aldehyde dehydrogenase (ALDH), expressed Hh ligands, which acted cell-autonomously to activate downstream Hh targets in a non-canonical, SMO-independent, PTCH1-dependent manner [124]. One further study suggested that Hh ligands can bind cell-autonomously to CDO expressed by CRC cells, and that CDO can act as a dependence receptor, uncoupled from its role as a Hh coreceptor [125]. Finally, another study showed that TGF-+beta 2, produced by stromal cells, can activate GLI2 in colon tumor cells [126], providing a further potential mechanism by which Hh ligands can act on CRC tumor cells. Figure 3 summarizes current models of both stromal and intra-epithelial effects of Hh signaling.

### 4.5. Failure of Hedgehog Inhibition in a Clinical Trial

In 2012, vismodegib, a small molecule inhibitor of human SMO was approved by the United States’ Food and Drug Administration (FDA) and later by the European Medical Agency (EMA), for the treatment of advanced BCC [127]. This approval opened up the interesting possibility of using Hh inhibition in combination with standard therapies in other solid cancers. For CRC, one phase II clinical trial involving 35 medical centers in the United States investigated the effect of adding vismodegib to standard chemotherapy in patients with metastatic CRC (mCRC) [128]. Patients with mCRC were randomized for treatment with either vismodegib or placebo in combination with standard chemotherapy. The standard therapy included a chemotherapy doublet with 5-fluoruracil, and either oxaliplatin or irinotecan, and the vascular endothelial growth factor (VEGF) antibody, bevacizumab. As the primary endpoint, the authors assessed progression-free survival (PFS) in a total of 199 patients, 96 of whom received vismodegib. The overall response rate was 51% (90% confidence interval: 43–60) for placebo-treated patients, and 46% (90% confidence interval: 37–55) in the vismodegib-treated group. Expression levels of *SMO*, *GLI1,* and *PTCH1* as assessed in tumor tissue by real-time quantitative polymerase chain reaction (qPCR) did not predict a response to the addition of vismodegib. In summary, the study showed no benefit from adding vismodegib to chemotherapy in patients with mCRC, and the authors ended on a discouraging note, suggesting that further testing of vismodegib in this context is not warranted [128]. Regrettably, the negative results from this CRC trial are in line with trials that evaluated Hh inhibitors in other types of solid tumors, most notably pancreatic ductal adenocarcinoma [129,130], small-cell lung cancer [131], and ovarian cancer [132]. The trial using Hh inhibitors in mCRC did not assess the degree of suppression of Hh targets, or of the duration of any Hh suppressive effect in the colon. It is conceivable that the dose of vismodegib used was not sufficient to suppress Hh signaling in the intestine, either in extent or over time, and it is possible that there are rebound effects with periods of increased Hh target gene expression [128]. Because inhibitor levels would have to be measured directly in the colon, these data are difficult to obtain from human patients. However, similar doses of vismodegib have shown robust clinical effects in the treatment of BCCs [127,128], and mouse models have provided evidence for a Hh suppressive effect of vismodegib in the colon, as measured by mRNA expression of *Gli1* [61,71], which argues against an insufficient dose used in the trial. Functionally, treatment with vismodegib has been shown to increase the number and size of colorectal tumors in mice in colitis-associated models [61,71], while the experimental Hh agonist, SAG21k, which acts at the level of SMO, decreased the colorectal tumor burden [71]. Together with the results from the trial, the data therefore clearly argue against a positive net effect of Hh inhibition in unselected mCRC patients.

## 5. Future Directions for Targeting Hedgehog Signaling in Colorectal Cancer

Although inhibiting Hh signaling in patients with mCRC had no beneficial effect, results from mouse models suggest that manipulating this important morphological pathway impacts tumor growth in the colon [61,66,71]. In contrast to the clinical trial, the use of Hh agonists rather than antagonists could be a way forward. However, in the context of the proposed cell-intrinsic, cell-autonomous role for non-canonical Hh activation in a subset of CRC cells, such an approach could have opposing effects in both epithelial and stromal compartments. Hence, a better understanding of the downstream targets of canonical Hh signaling in the stroma will be crucial for more refined, targeted therapies of stromal cells. Recent single-cell analyses of stromal cells in healthy colon have revealed a striking complexity of cellular states, and found that Hh-active cells, as defined for example by the expression of *Gli1*, represent a heterogenous group, and express different molecules that can modify epithelial differentiation and proliferation via distinct signaling pathways, including Wnt and BMP [49,81]. At the very least, it will be important to answer two major questions about the heterogeneity of Hh-active cells in the tumor stroma in order to gain a better understanding of the functional role of Hh in CRC:

(1) Where in the stroma are the cells expressing Hh targets located in relation to the tumor cells? Analyzing the non-random proximity of a given stroma cell type to tumor cells in a specific state should help to understand juxtacrine or paracrine signaling mechanisms. For example, are tumor cells receptive to Wnt or BMP ligands secreted by stromal cells? Are juxtacrine signaling processes in place between those two compartments, which are normally separated by the basement membrane in healthy colon? Are stromal cells capable of modifying the differentiation or proliferation of adjacent tumor cells? In-depth analyses of tumor heterogeneity that take into account both tumor and stroma should help to answer these questions.

(2) Which cells are receptive to modifications of the canonical Hh pathway, i.e., which of the cell types or states depend on Hh signaling, and in which can the activation or inhibition of Hh induce significant, functionally important changes? As gene expression of Hh downstream targets is highly dependent on cellular states, tissue context, and ligand dosage [133,134], a more detailed understanding of the role of Hh in each of these cell states, and in the context of their spatial location to tumor cells should reveal targets that allow a specific manipulation of unfavorable stroma–tumor interactions, while leaving beneficial, anti-tumoral interactions intact. It is not unlikely that roles for Hh beyond paracine, direct stroma–tumor interactions emerge upon closer investigation; for example, Hh plays a role for angiogenesis and lymphangiogenesis in several organs [135,136], although the importance of Hh for vascular remodeling in CRC needs to be better defined. Towards disentangling the role of Hh in different stromal cell types, modern single-cell based techniques will likely provide valuable insights. On a clinically more applicable level, it will be important to relate Hh activity to CRC subtypes (e.g., CMS1-4) and thereby identify patients with tumors susceptible to Hh manipulation.

Aside from mutations in Wnt and KRAS pathways, genetic loss of *SMAD4,* a central component of the SMAD complex that has DNA binding activity, occurs in 20–40% of CRCs, [137]. SMAD4 is required for the intracellular translation of external TGF-beta and BMP signals [138]. The loss of *SMAD4* alters the response of CRC cells to BMP agonists, such that invasion and migration increase, while the degree of differentiation decreases in cells lacking *SMAD4* [138]. The loss of *SMAD4* is particularly frequent in mCRC [139], the patient cohort studied in the vismodegib phase II clinical trial [128]. Congruent data suggest that Hh-responsive stromal cells are pivotal modifiers of the BMP signal that reaches epithelial or cancer cells [61,62,140]. Given the diametrical change in the response to BMP signals following the loss of *SMAD4* [138], it will be interesting to investigate the response of tumor cells to stromal Hh signaling in CRCs which retain *SMAD4* versus those that have lost *SMAD4* expression.

The release and transport of intestinal morphogens, such as Hh, are regulated on several levels. As outlined in the first section, HSPGs, such as glypicans and syndecans, play central roles in this context, and many are differentially expressed in CRC compared to normal mucosa (reviewed in [15]). Given the general roles of HSPGs, changes in their expression are likely to impact a multitude of signaling pathways involved with colonic carcinogenesis, and it will be important to decipher the individual contributions of HSPGs in detail, as they represent potentially druggable targets for future therapies.

Several studies in mice have suggested an important role for Hh-responsive stromal cells in modifying the intestinal immune response [62,70,71,72], and mice with inflammation-induced tumors were protected from polyp formation by Hh agonist treatment [61,71]. For patients with mCRC, characterized by high levels of MSI, immunotherapy with an anti-PDL1-antibody doubled the two-year survival rate from 18% to 37% compared to standard chemotherapy [141]. While immunotherapy has finally become a first-line option for a defined group of CRC patients, the majority of CRCs are microsatellite stable (MSS), and thus are not candidates for current immunotherapy regimens [95]. Given the diverse immunomodulatory effects of Hh signaling via stromal cell cytokine and chemokine release, a deeper understanding of the impact of Hh signaling modifications on immune checkpoints in both the CRC stroma and the cancer cell compartment is warranted (reviewed in [142]).

Finally, CRC mortality is caused by metastatic disease, rather than by the primary tumor in the colon [82]. Most commonly, distant CRC metastases are found in the liver and lung [82]. In these organs, cancer cells meet novel interaction partners including epithelial cell types such as cholangiocytes or pneumocytes, specialized stromal cells, and a distinct immune cell infiltrate. Currently, we are basing our predictions of the functional role of Hh signaling modifications largely on research in animal models that recapitulate early in situ tumor stages in the colonic mucosa. Given the clinical importance of metastatic disease and the fact that clinical trials of novel treatments in CRC, such as the phase II vismodegib trial [128], primarily include patients with metastases, it will be particularly important to improve our knowledge of Hh signaling in the stroma of metastatic tumor sites. Interestingly, stromal Hh signaling has documented functional roles in both major CRC metastatic sites: in the liver, Hh acts in a pro-fibrotic manner and is required for liver repair after injury [143,144], while in the lung, it is likely that activated stromal Hh signaling plays a general role in fibrosis development across different fibrotic lung diseases (reviewed in [145]). These data show the ability of Hh-responsive cells in the liver and lung to react to injury. In the future, it will be important to decipher the responses of these cells to injury induced by metastasized tumor cells, and to gain a better understanding of the similarities in their responses to the diverse stromal cells of the colon. In CRC liver metastases, the presence of a stromal capsule at the metastatic invasion front is associated with a favorable outcome [146,147]. The signaling pathways that underlie the development of this clinically important stromal border between tumor cells and the liver parenchyma are only superficially understood, and based on the role of Hh signaling in liver fibrosis, gaining a clearer picture of Hh expression at the invasion front of liver metastases is likely to be clinically relevant.

## 6. Concluding Remarks

Hh signaling in the colon is predominantly paracrine, directed from the epithelium to the stromal cells. It is likely that this feedback loop also exists in CRC, where the sum of Hh-driven stromal signals has pro-differentiating, anti-proliferative effects. Recent studies in mice have shown that, at least in the healthy colonic crypt, a subset of Hh-responsive cells provides essential Wnt ligands that uphold the intestinal stem cell state in the crypt. At the crypt apex, GLI1-positive cells secrete BMP ligands that induce the differentiation of enterocytes. Hence, Hh-responsive cells in the normal colonic stroma seem to play opposing roles and can have pro- or anti-differentiating effects on adjacent epithelial cells. In addition, Hh-responsive stromal cells play a vital immune-modulatory role in the colon, the details of which are not well understood in human disease. Together with data that support a role for non-canonical Hh signaling in CRC cancer cells, a complex picture of Hh signaling in the colon is emerging, which may explain the difficulties of demonstrating an effect of Hh inhibition in the clinical setting. Nevertheless, the data suggest an important role for Hh in CRC and demonstrate that it will be necessary to disentangle the diverse roles of Hh in different cellular compartments and in the spatial relationships of the various cell types. Recent technical advances in single cell sequencing methods, spatial transcriptomics and multiplex immunofluorescence techniques provide exciting opportunities for future studies and inject hope that Hh might yet become a target for future therapies in CRC and other solid tumors.

## Figures and Tables

**Figure 1 ijms-22-01025-f001:**
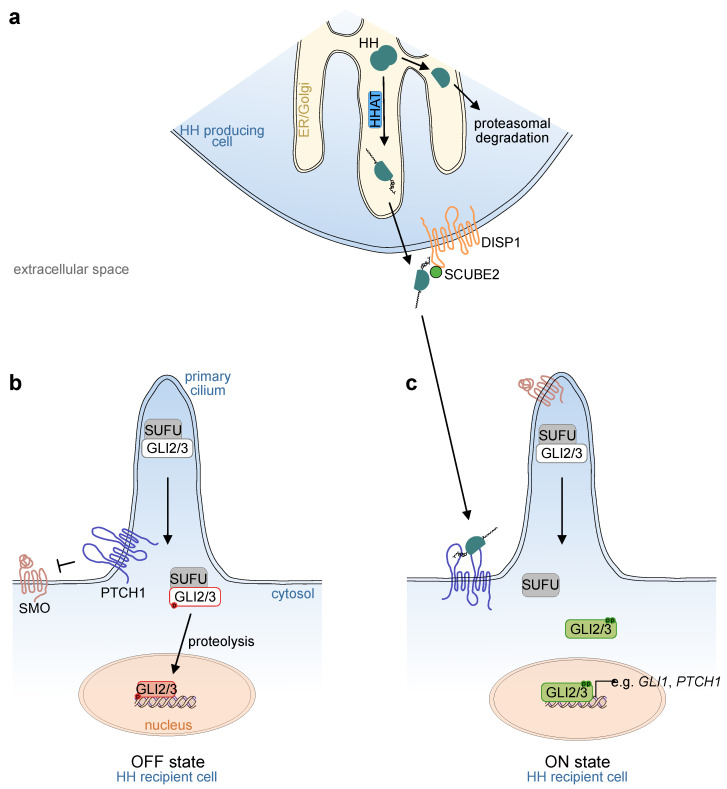
Essential molecules of the mammalian Hedgehog signaling pathway and their interactions. (**a**) In the producing cell, Hedgehog (Hh) ligands are formed as precursor proteins and undergo posttranslational modification within the endoplasmatic reticulum (ER) and Golgi apparatus. The C-terminus is cleaved and degraded by the proteasome. The former N-terminus is palmitoylated and conjugated with a cholesterol moiety by Hh acetyltransferase (HHAT), as well as autocatalytically. Secretion is facilitated by cell membrane-residing Dispatched 1 (DISP1) and its cofactor SCUBE2. (**b**) In the absence of Hh ligands, Patched 1 (PTCH1) prevents Smoothened (SMO) from entering the primary cilium. Full-length Glioma-associated oncogenes 2 and 3 (GLI2/3) transcription factors accumulate within the primary cilium in a complex with Suppressor of fused (SUFU). GLI2/3 factors are phosphorylated at their repressor domains, triggering proteolytical cleavage into truncated repressor proteins, which can enter the nucleus and inhibit the expression of Hh target genes (GLI2/3 in red). (**c**) Hh ligands interact with PTCH1, which releases repression of SMO. SMO accumulates at the top of the primary cilium and full-length GLI2/3 is released from its inhibitor SUFU. GLI2/3 are further phosphorylated at their activator domains. The activated proteins (GLI2/3 in green) enter the nucleus and activate the expression of Hh target genes, including *GLI1* and *PTCH1*.

**Figure 2 ijms-22-01025-f002:**
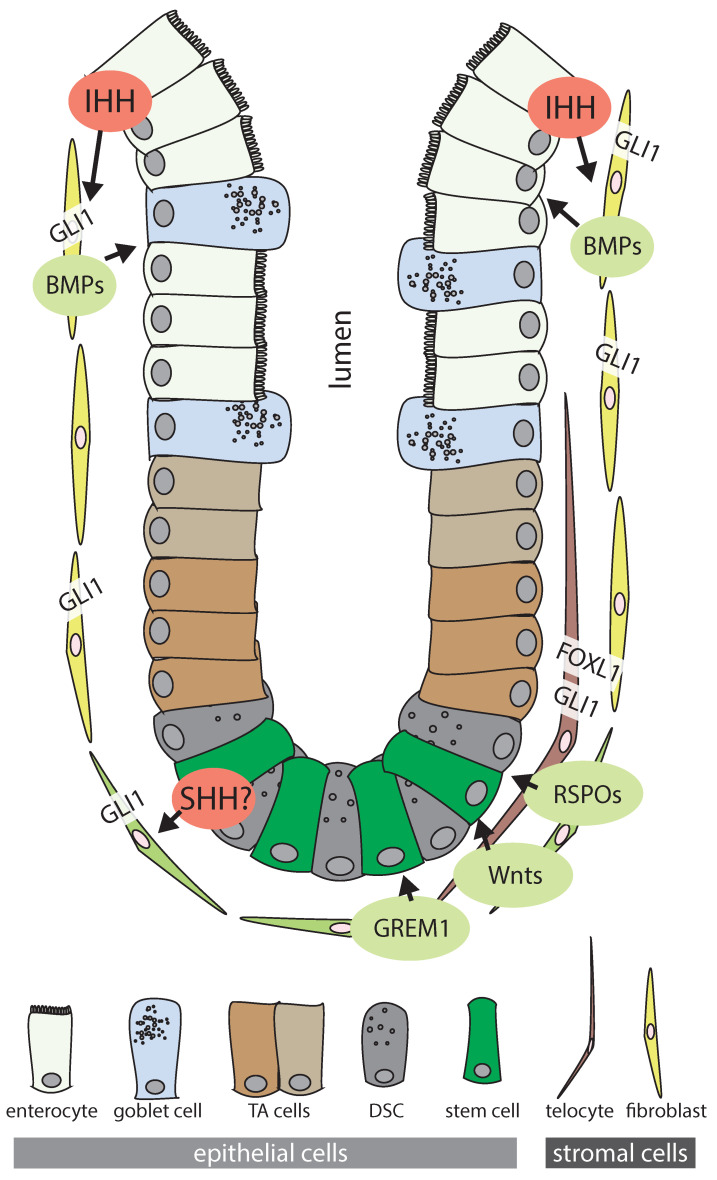
The cell types and signaling molecules involved in colonic Hedgehog (Hh) signaling. Cells with active Hh signaling are located in the stroma around the colonic crypt and comprise different types of fibroblast-like cells, as well as telocytes. In the epithelial layer, stem cells (green) are located at the base of the crypt, adjacent to deep crypt secretory cells (DSCs) [68]. Epithelial cells follow differentiation trajectories via transit-amplifying (TA) cells that end in enterocytes or goblet cells. For the sake of clarity, other cell types, such as tuft cells or enteroendocrine cells are not shown. Enterocytes towards the luminal side secrete Hh ligand, largely IHH. Stromal cells reacting to the Hh ligand express GLI1 and respond with secreting differentiation factors such as BMP ligands, mainly at the luminal part and close to differentiated epithelial cells. In the crypt region, stromal cells provide both Wnt ligands and BMP inhibitors, such as GREM1, that maintain the epithelial stem cell state. Telocytes can compartmentalize the secretion of both differentiation factors and stem cell factors. Transcription factors (GLI1, FOXL1) expressed in stromal cells are shown in the rectangles. Secreted molecules in are shown in the circles. The legend of cell types is shown at the bottom of the figure.

**Figure 3 ijms-22-01025-f003:**
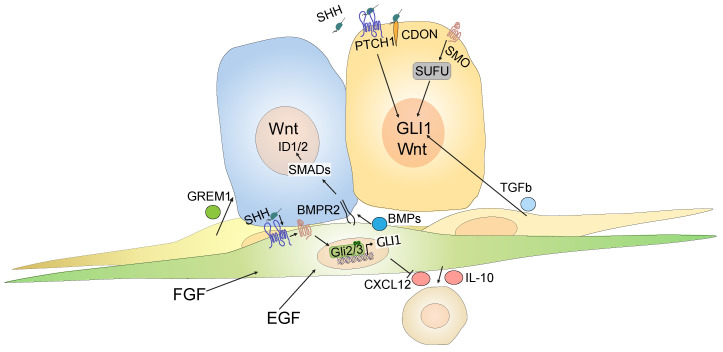
Epithelial vs. stromal Hedgehog signaling in colorectal cancer. Most tumor cells are characterized by high Wnt signaling levels, but some may produce Hh ligands (blue cell), e.g., SHH. A Hh responsive stromal cell (green) receives the ligand via PTCH1 and SMO (“canonical” Hh signaling). Most stromal cells (in the background) do not respond to Hh ligands. The receptive stromal cells respond by secreting, e.g., BMP ligands, which act via BMP receptors (e.g., BMPR2) and SMADs on the tumor cell side to activate pro-differentiating ID1 and ID2; conversely, tumor cells are exposed to BMP inhibitors such as GREM1, secreted by other stromal cells. In addition, some GLI1-expressing cells secret immunomodulating factors, such as IL-10, or seize secreting CXCL12, possibly impacting on inflammatory cells of the tumor microenvironment. Hh signaling strength can independently be modified by other factors, such as EGF and FGF, although their influence on the stromal Hh response in cancer is less clear than that in development. Other tumor cells (yellow cell) may exhibit non-canonical activation of the GLI code, e.g., PTCH1-dependent, SMO-independent, directly via SMO, via a Wnt–SUFU-GLI axis, or in response to TGFB2 produced by stromal cells. In addition, Hh ligands can act in an autocrine way via CDON/BOC as a dependence signal.

## Data Availability

Not applicable.

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
