# Peer review of "Hedgehog Signaling in Colorectal Cancer: All in the Stroma?"

_ijms, 2021, doi:10.3390/ijms22031025_

Round 1

Reviewer 1 Report

I appreciate the opportunity to review this manuscript on „ Hedgehog Signaling in Colorectal Cancer: All in the Stroma?”

The development of CRC is a complicated process involving various mechanisms and factors, and requires a complex of network among different signal transductions. The Hh signaling pathway is also involved in the signaling transductions of the tumorigenesis of CRC.

The Hh signaling pathway was shown previously to be multi-functional in the formation of CRC. The involvement of the Hh pathway in gene mutation, metastasis, apoptosis, and angiogenesis was discovered.

Activation of the Hh signaling pathway is also reported to be associated with Gli1-induced lymphangiogenesis and tumor cell regeneration in CRC, which is correlated to the metastatic ability and drug resistance in chemotherapy of CRCs.

In many studies, downregulation of Ihh has been observed as an early event in the formation of CRC. Van den Brink et al found that loss of Ihh expression precedes the development of dysplasia in colon carcinogenesis.

It is believed that overexpression of Shh and its downstream components is highly correlated with the formation and metastasis of CRCs, whereas it’s detailed mechanism remains unclear.

Overall:

This Review article is well written and provide valuable information to surgeons and oncologists.

Author Response

The development of CRC is a complicated process involving various mechanisms and factors, and requires a complex of network among different signal transductions. The Hh signaling pathway is also involved in the signaling transductions of the tumorigenesis of CRC.

We agree with the reviewer that CRC carcinogenesis is complex. In line also with comments from reviewer #2, we have expanded the section on “Epidemiology and genetic background” (4.1.) to reflect this. We touch upon intraepithelial Hh signaling and its role in CRC development in section 4.4., and we feel that, apart from more specific additions detailed below, further additions would be not warranted for the sake of focus on the microenvironment, as intended for this Special Issue of the journal. 

The Hh signaling pathway was shown previously to be multi-functional in the formation of CRC. The involvement of the Hh pathway in gene mutation, metastasis, apoptosis, and angiogenesis was discovered.

Activation of the Hh signaling pathway is also reported to be associated with Gli1-induced lymphangiogenesis and tumor cell regeneration in CRC, which is correlated to the metastatic ability

In answer to the two above comments: We fully agree that Hh has multifunctional roles in CRC. We argue that the canonical pathway is largely inactive in CRC cells in vivo, as supported by mouse models and independent RNA-seq datasets of human CRC. However, we discuss the roles of non-canonical Hh activation in CRC (epithelial) cells in section 4.4. We appreciate the reviewer’s comment on the complex roles of Hh activation in epithelial cells, and we have added a statement on the possible roles of Hh in angiogenesis and lymphangiogenesis (section 5), which are processes that take place in the microenvironment. For the sake of the intended focus on the stromal roles of canonical Hh signaling, we limited ourselves to these textual additions, but added a reference to a more recent review on these important aspects for the readers convenience (section 4.4., on of first paragraph).

In many studies, downregulation of Ihh has been observed as an early event in the formation of CRC. Van den Brink et al found that loss of Ihh expression precedes the development of dysplasia in colon carcinogenesis.

In the revised manuscript, we discuss the aspect of Hh dysregulation in early tumor stages (section 4.4.). More recent data from the van den Brink lab (Büller, Gastroenterology 2015), suggested upregulation of IHH mRNA in adenomas, not fully congruent with the initial findings.

It is believed that overexpression of Shh and its downstream components is highly correlated with the formation and metastasis of CRCs, whereas it’s detailed mechanism remains unclear.

As the involvement of the stroma in this context is unclear, we would suggest omitting a deeper discussion of the role of Shh overexpression for CRC metastases in order to help focus on the paracrine signaling pathway interactions.

Overall:

This Review article is well written and provide valuable information to surgeons and oncologists.

We are glad to learn about the overall positive assessment of our manuscript.

Reviewer 2 Report

This is a quite meaningful summary of our knowledge on the SHH signalings and its role in colorectal cancer.

Corrections to be made:

p2/L64: glypican is not the only transmembrane HSPG but major one is syndecane (1/2 is involved in colorectal carcinogenesis as loss) or CD44v3/v6. In ECM none can be found only decorin (CSPG) and perlecane (HSPG)

p2/L79: cilia formation. It is not clear how cilia formation come to intestinal epithelium which is not ciliated….

p10L412: colorectal carcinogenesis: it turned out that one of the major etiological factor for CRC carcinogenesis is genotoxic E.Coli infection (Plequezuelos-Martinez C et al. Nature 580:269,2020)

p10L426: NRAS is also oncogenic driver of CRC (5-10%) and RAS signaling pathway alterations (RAS and BRAF mutations) characterize the majority of CRC (~60%). APC plays a role as the major oncosuppressor…of CRC.

Going closer to the issue of SHH signaling in CRC: it is important to consider the 4 consensus molecular signatures of CRC: CMS1-4.

CMS1: is the MSI form which contains the BRAF mutants

CMS2: is the CIN-WNT mutated form

CMS3: is the RAS mutated version

and CMS4: is the mesenchymal form.

Based on this it is evident that SHH signaling can be involved in CMS2 or CMS4 versions…but not in all molecular forms. Concerning the SHH inhibitor issue major problem is the non-targeted clinical use in CRC. APC mutation was not used as predictive marker nor was the RAS signaling pathway mutations analysed as possible negative or positive predictive factor…

When one look for possible trivial use of SHH inhibitor in CRC the heteriditary form FAP could well be an option…

Author Response

This is a quite meaningful summary of our knowledge on the SHH signalings and its role in colorectal cancer.

We would like to thank the reviewer for their critical evaluation of our review and are glad about the positive assessment.

Corrections to be made:

p2/L64: glypican is not the only transmembrane HSPG but major one is syndecane (1/2 is involved in colorectal carcinogenesis as loss) or CD44v3/v6. In ECM none can be found only decorin (CSPG) and perlecane (HSPG)

We have amended the introduction to include more detailed information on the HSPGs and their expression/localization in the different cellular and acellular compartments in the suggested section. In addition, we briefly discuss their potential role in colorectal carcinogenesis in “Future directions…”, section 5.

p2/L79: cilia formation. It is not clear how cilia formation come to intestinal epithelium which is not ciliated….

Thank you for this valuable remark. Indeed, primary cilia are largely absent in intestinal epithelial cells, in line with the paracrine wiring of Hh signaling. We added a comment on these data in the revised manuscript, section 2.2.

p10L412: colorectal carcinogenesis: it turned out that one of the major etiological factor for CRC carcinogenesis is genotoxic E.Coli infection (Plequezuelos-Martinez C et al. Nature 580:269,2020)

We have added a reference to this recent paper in section 4.1., Epidemiology and genetic background of colorectal cancer.

p10L426: NRAS is also oncogenic driver of CRC (5-10%) and RAS signaling pathway alterations (RAS and BRAF mutations) characterize the majority of CRC (~60%). APC plays a role as the major oncosuppressor…of CRC.

We have included NRAS mutations in section 4.1. and clarified that APC acts as a tumor suppressor in the same section.

Going closer to the issue of SHH signaling in CRC: it is important to consider the 4 consensus molecular signatures of CRC: CMS1-4.

CMS1: is the MSI form which contains the BRAF mutants

CMS2: is the CIN-WNT mutated form

CMS3: is the RAS mutated version

and CMS4: is the mesenchymal form.

Based on this it is evident that SHH signaling can be involved in CMS2 or CMS4 versions…but not in all molecular forms. Concerning the SHH inhibitor issue major problem is the non-targeted clinical use in CRC. APC mutation was not used as predictive marker nor was the RAS signaling pathway mutations analysed as possible negative or positive predictive factor…

We agree with the reviewer that it is important to consider the CMS classification. Indeed, our own unpublished data indicate that the Hh gene expression signature is higher in CMS2 and 4, compared to CMS1 and 3. However, we feel that we cannot yet exclude a role for Hh in the other subtypes based on available data. In the revised manuscript, we now introduce the CMS classification in section 4.1., and we have added a brief discussion of the potential value of CMS to better understand Hh signaling in CRC in “Future directions…”, section 5.

When one look for possible trivial use of SHH inhibitor in CRC the heteriditary form FAP could well be an option…

We agree that this genetically distinct form of CRC is likely to benefit from interference with Hh signaling, for example for tumor prevention. However, the mouse data (Büller et al., Gastroenterology 2015, and Gerling et al., Nature Comms 2016) point to the possibility that colonic and small intestinal tumors may have different requirements for Hh signaling, and we feel that this issue needs to be better understood before making more specific suggestions for FAP patients.